# Improving Stuttering Through Augmented Multisensory Feedback Stimulation

**DOI:** 10.3390/brainsci15030246

**Published:** 2025-02-25

**Authors:** Giovanni Muscarà, Alessandra Vergallito, Valentina Letorio, Gaia Iannaccone, Martina Giardini, Elena Randaccio, Camilla Scaramuzza, Cristina Russo, Maria Giovanna Scarale, Jubin Abutalebi

**Affiliations:** 1Vivavoce Medical Center, 20124 Milan, Italy; g.muscara@vivavoceinstitute.com (G.M.);; 2Vivavoce Research Department, 20124 Milan, Italy; 3Department of Medicine and Aging Sciences, University “G. d’Annunzio” of Chieti–Pescara, 66013 Chieti, Italy; 4University Centre of Statistics in the Biomedical Sciences (CUSSB), Vita-Salute San Raffaele University, 20132 Milan, Italy; scarale.maria@unisr.it; 5Centre for Neurolinguistics and Psycholinguistics, Vita-Salute San Raffaele University, 20132 Milan, Italy; 6C-LaBL, UiT-The Arctic University of Norway, 9019 Tromsø, Norway; 7CHIC, Higher School of Economics (HSE University), 101000 Moscow, Russia

**Keywords:** stuttering, sensory feedback, personalized intervention, fluency enhancement, DIVA model

## Abstract

Background/Objectives: Stuttering is a speech disorder involving fluency disruptions like repetitions, prolongations, and blockages, often leading to emotional distress and social withdrawal. Here, we present Augmented Multisensory Feedback Stimulation (AMFS), a novel personalized intervention to improve speech fluency in people who stutter (PWS). AMFS includes a five-day intensive phase aiming at acquiring new skills, plus a reinforcement phase designed to facilitate the transfer of these skills across different contexts and their automatization into effortless behaviors. The concept of our intervention derives from the prediction of the neurocomputational model Directions into Velocities of Articulators (DIVA). The treatment applies dynamic multisensory stimulation to disrupt PWS’ maladaptive over-reliance on sensory feedback mechanisms, promoting the emergence of participants’ *natural voices*. Methods: Forty-six PWS and a control group, including twenty-four non-stuttering individuals, participated in this study. Stuttering severity and physiological measures, such as heart rate and electromyographic activity, were recorded before and after the intensive phase and during the reinforcement stage in the PWS but only once in the controls. Results: The results showed a significant reduction in stuttering severity at the end of the intensive phase, which was maintained during the reinforcement training. Crucially, worse performance was found in PWS than in the controls at baseline but not after the intervention. In the PWS, physiological signals showed a reduction in activity during the training phases compared to baseline. Conclusions: Our findings show that AMFS provides a promising approach to enhancing speech fluency. Future studies should clarify the mechanisms underlying such intervention and assess whether effects persist after the treatment conclusion.

## 1. Introduction

Stuttering is a speech disorder in which individuals know what they want to say. Still, verbal output is disrupted by involuntary (i) repetition of sounds, syllables, or words; (ii) sound prolongation; and (iii) pauses or blocks. These primary symptoms are typically accompanied by secondary behaviors, such as involuntary physical movements (e.g., eye-blinking, jaw jerking, and movements of the head, arm, or other body district) [1,2].

Stuttering typically emerges during childhood (developmental stuttering) [3,4], with an incidence of up to 10% [5,6]. Although most children recover spontaneously within a few years from the onset, around 1–2% do not, and individuals present stuttering throughout their lives [7], with a male-to-female ratio of 4 to 1 [6,8]. Stuttering can also emerge during adulthood (acquired stuttering), typically following a neurological disease or psychological trauma or as a side effect of certain medications (see, for recent works, [9,10,11]). Of note, stuttering behaviors and severity vary between and within individuals; indeed, they may change over time depending on situational, emotional, and linguistic factors [12,13].

Although the essential feature of stuttering is a disturbance in the typical fluency and time patterning of speech [14], converging evidence suggests that stuttering is a complex condition that goes beyond verbal communication and includes cognitive, social, emotional, and psychological domains [5,15]. The disorder often includes stuttering anticipation, negative thoughts regarding themselves, more frequent experiences of negative emotions such as anxiety and shame compared to non-stuttering individuals, and maladaptive coping strategies such as avoidant behavior and social withdrawal [5,16,17,18,19]. These elements represent both consequences and potential factors maintaining stuttering, creating a vicious cycle that amplifies the disorder’s impact on people who stutter (PWS) lives. In line with such complexity, stuttering severity shows a negative correlation with education attainment [20,21] and occupational outcomes, with stuttering children repeating a grade with a higher percentage compared to control peers [22,23] and adults reporting fewer financial incomes and employment rates than non-stuttering individuals [24,25]. Considering the complexity of the disorder and its significant social and psychological impact on individuals, it is essential to develop evidence-based, cost-effective treatments.

### 1.1. Augmented Multisensory Feedback Stimulation: Theoretical Background

Converging evidence supports the idea that the core of stuttering is characterized by dysfunctions in speech–motor control [15,26], the properties and functioning of which have been simulated through different computational models (see [27] for a recent overview comparing models’ architectures and features). In the current work, we focus on one of these models, namely the Directions into Velocities of Articulators (DIVA) [28,29] since it has often been applied to stuttering [30,31] and is deeply related to our intervention.

In detail, the DIVA model is a neurocomputational model of speech motor control that integrates two key components, namely internal models and control systems, to explain how we produce speech sounds. Internal models are sensorimotor maps representing the movements of articulators, such as the tongue, lips, and jaw, and their perceptual consequences, such as the sounds produced. These maps create an internal representation, including perceptual goals for producing speech sounds. Each sound has a perceptual target (auditory and somatosensory goals) maintained in the brain, and speech production involves reaching this target. According to DIVA, internal models are built mainly during infancy and early childhood. They are further developed through the interaction of motor learning, sensory feedback, and environmental influences (like hearing and imitating speech), allowing continuous updates and refinement of our speech sound production.

The second key element of DIVA concerns integrating feedforward and feedback control mechanisms to produce self-generated, appropriate, and goal-directed speech outcomes. The feedforward control system sends pre-planned motor commands to the articulators without relying on immediate sensory feedback. This system relies on internal models representing the correct way to produce sounds. The feedforward control is used during fluent speech, such as during conversation, where real-time corrections or adjustments are not required, enabling quick and efficient speech production. On the other hand, it lacks flexibility and the possibility of correcting errors. The feedback control system comprises two modality-specific mechanisms: one for auditory feedback (i.e., the sound of an individual’s voice) and one for somatosensory feedback (i.e., kinesthetic or proprioceptive information from muscle spindles or the tongue, lips, and palate). The feedback controller monitors the sensory outcomes of speech by continuously comparing the feedback to the desired result. Any discrepancy, such as the mispronunciation of a sound, is used to generate new motor commands and make real-time adjustments to correct errors or improve accuracy. As a limitation, the feedback mechanism is slow because of its nature and does not account for rapid movements [30].

According to DIVA, feedforward and feedback controllers remain active throughout the individuals’ lifespan, even though their balance may shift over time. In earlier phases, speech development relies exclusively on feedback control, particularly auditory feedback control. Sensory feedback trains the feedforward mechanism, in which map nodes encoding motor, aural, and somatosensory output trajectory are learned. Once the internal models become reliable, a shift in the balance between the controllers takes place, and the feedforward system becomes dominant in speech motor control. The feedback controllers, however, remain active in introducing corrective motor commands whenever errors are detected.

Max and colleagues [31] applied the DIVA principles to stuttering, suggesting two possible explanations for the neurophysiological basis of the disorder. The first hypothesis suggests that PWS would have unstable or insufficiently activated internal models due to difficulties: (i) in their acquisition and updating or (ii) in their activation and use. According to this hypothesis, inaccuracies or instability of internal models would cause mismatches between predicted and actual sensory feedback, especially the auditory feedback, thus prompting corrective motor adjustments. The online corrections are carried out by the feedback controller, which, as aforementioned, is slower than the feedforward one, leading to repetitions or prolongations in speech, representing the so-called overt symptoms of stuttering. According to the internal model hypothesis of impairment, namely a deficit in the acquisition and updating vs. activation and use of internal models, behavioral evidence seems to support the latter. Indeed, most PWS sometimes experience their speech being fluent and free from interruptions, blocks, or repetitions [12,13]. A deficit at the acquisition/updating level would predict a constant malfunction in the system. Conversely, the situational nature of stuttering implies an impairment in the activation and use of internal models [32].

The second hypothesis proposed by Max and colleagues suggests that the difficulties experienced by PWS would not entirely be at the level of internal models. Rather, the authors posit a weakness of the feedforward control system, leading to an over-reliance on feedback control for speech production (see [30] for a detailed discussion of different theories and their predictions). When movements depend primarily on afferent feedback control, delays in sensory signal processing, i.e., the time associated with detecting the mismatch between the intended output and the sensory feedback [33,34], could destabilize the system. These instabilities, especially during rapid movements, may result in effector oscillations and system resets, leading to observable speech dysfluencies typical of stuttering.

In the context of this hypothesis, Max and colleagues suggested that PWS prefer longer movement durations during speech and nonspeech movements, as these are less susceptible to processing delays than faster movements. In our clinical experience, we do not observe a marked slowing of speech rate as a prototypical characteristic of PWS. Instead, in line with others [35,36,37,38], we detect a significant increase in the *effort* associated with speech production. Such effort has been conceptualized as primarily cognitive by the Vicious Circle Hypothesis [35], according to which PWS would deploy excessive resources in monitoring their speech production system. Conversely, other researchers emphasized that fluent speech is effortless, considering the little cognitive preparation and small amount of muscular exertion required. For instance, Conture and Kelly [37] suggest that the physical concomitants observed during stuttering may reflect an ‘overflow’ due to the physical, mental, and emotional effort involved in ‘getting the word out’ when PWS want to start or continue to speak. Although objective instruments measuring muscular activity are available, studies involving electromyographic registration across orofacial, jaw, laryngeal, and respiratory muscles reported inconsistent results [39,40,41]. This aligns with previous insights suggesting that effort is a subjective rather than an objective measure, leading to issues in measurement definition, reliability, and validity [42]. The role of effort plays a central role in our intervention, and we will return to this point below in the presentation of our intervention.

The DIVA framework has significant consequences considering the role of perturbating the feedback controllers to enhance fluency in PWS and, therefore, to develop interventions to improve stuttering behaviors. In this vein, perturbation of the auditory feedback during speech temporarily increases the fluency of PWS (e.g., [43]). Techniques such as choral reading, metronome-timed speech, and the use of electronic devices to manipulate PWS’ voice by masking, delaying, and altering its frequency (known as Altered Auditory Feedback interventions, AAF [43]) have been found to reduce stuttering and increase speech fluency. Although auditory feedback has received more attention [30], visual and vibrotactile feedback has also been investigated. As for visual feedback, some auditory techniques can be ‘translated’ into visual modality through visual rhythm delivery, for instance, by using flashing lights [44] or, more efficiently, through visual choral speech (i.e., a second speaker mimics speech without auditory signals) [45]. Moreover, visual feedback improves stuttering when showing the speaker’s face or face portions to PWS synchronously or asynchronously [46,47]. Lastly, vibrotactile feedback has been manipulated to modulate speech fluency. Studies suggested that stuttering is reduced when participants are asked to feel the thyroid cartilage vibration with their fingers [48] or through mechanical devices (e.g., [49], independently from the vibrator positions (i.e., hand, forehead, and sternum) [50].

These techniques and their underlying concepts, however, can also be effectively embedded in other models, such as those considering stuttering as a disorder of timing [51,52,53]. Indeed, behavioral and neuroimaging data have supported timing difficulties in PWS and suggest a primary impairment in the cortico-basal ganglia–thalamocortical loop, which is involved in motor initiation [54,55] and, consequently, in initiating speech movements [56,57]. Speech motor control requires the coordination of complex, interconnected motor sequences that must be precisely timed. At the same time, motor sequences should be promptly adjusted in response to changes in speech rate, emotional state, rhythm, intonation, and prosody as the speech flow continues uninterrupted. External timing cues, such as in choral reading, metronome-timed speech, and the previously mentioned techniques, typically reduce or eliminate stuttering [5]. These methods, however, are not without any weaknesses: the effects are temporary and disappear when individuals adapt to the perturbation or when the alteration of feedback ends. Follow-up evaluations have been rarely conducted, limiting our understanding of the long-term impact of these interventions on PWS’ lives [58] Moreover, the techniques are considered weak from an ecological perspective in real-life contexts where sensory feedback involves multiple sources and can be noisier [30,59].

### 1.2. Augmented Multisensory Feedback Stimulation: A New Approach to Improve Stuttering

Augmented Multisensory Feedback Stimulation (AMFS), developed from a decade of experience treating PWS, aims to restore an individual’s natural fluent voice, characterized by minimal disfluency, regular utterance rate, natural-sounding speech quality, and little effort [60]. As mentioned above, our fundamental assumptions are that (i) internal models are intact in PWS but become unstable when perturbed, which would explain why PWS experience moments of fluent speech, and (ii) PWS often exert excessive effort during phonatory acts. Frequent experiences of instability during PWS’ childhood influence the speech–motor system, leading to an over-reliance on feedback control and forcing the feedforward control system to send new commands by adding more significant effort in terms of cognitive resources and physical activation. The recurrence of such events establishes a *stuttering response*, namely an involuntary and maladaptive motor behavior. Like any motor behavior, its repetition strengthens it and makes it more automatic so that, once consolidated, the stuttering response is difficult to alter or eliminate [15]. From these considerations, AMFS aims to (i) disengage the automaticity of the overlearned response of stuttering; (ii) achieve the desired motor target, namely PWS’ natural fluent voice; (iii) consolidate the motor plans necessary for the production of the natural voice, promoting and reinforcing re-engagement with the new target to create an automatic alternative strategy over the stuttering response; and (iv) foster system flexibility to perturbation, thus generalizing the alternative *fluent response* across different conditions.

In AMFS, the acoustic, visual, somatosensory, and proprioceptive signals are continuously manipulated, preventing individuals from adapting to the afferent inputs. As a result, we speculate that the typical over-reliance of PWS on feedback is disrupted, and probably, their speech motor control system is overloaded due to the multiple sources of stimulation, leading to a ‘functional breakdown’. In this phase, their motor system can no longer rely on sensory feedback, and, in any case, it does not have the necessary resources to do so. Therefore, the produced output is an alternative and less effortful strategy, namely a fluent response. After pushing participants to achieve this goal, the training aims at reinforcing and consolidating the motor behaviors necessary to produce the fluent response and generalizing it across different contexts, including those characterized by extreme perturbations and highly challenging conditions, such as those used in AMFS.

At an operational level, the training program includes an intensive phase followed by a reinforcement phase. The intensive phase lasts thirty-two hours, spread over five consecutive workdays. The reinforcement phase is individually tailored regarding session frequency and duration. It typically includes at least one remote weekly session of half an hour plus one monthly in-person meeting of four hours for six to nine months. While the intensive training is primarily focused on the first three aims (disengagement from the stuttering response, achieving individuals’ fluent voice, and promoting re-engagement with the fluent response), the reinforcement phase aims to stabilize and strengthen the fluent response and generalize its implementation across different situations, especially the ones feared and avoided.

The AMFS training comprises three main activities: (i) auditory stimulation, (ii) visual stimulation, and (iii) visual plus somatosensory stimulation (Figure 1 shows some pieces of equipment). The technology employed in this study is protected by a patent (patent number: 102022000021813, Italian Patent and Trademark Office, De Vita & Muscarà, 2024).

During auditory stimulation, participants are placed in front of a monitor while wearing headphones that return their voice, manipulated according to multiple parameters, including frequency, delay, and volume. Manipulation is provided through an equalization system. Unlike other interventions, such as AAF, our approach continuously manipulates the auditory stimulus, which is not simply delayed or altered; instead, it is constantly slowed, accelerated, and exaggerated, thus preventing motor system adaptation and favoring functional breakdown. Participants are asked to focus on their speech production, which includes spontaneous speech, for instance, introducing themselves and reading aloud from provided texts. The trainers encourage participants to continue speaking/reading despite the auditory manipulation, changing their pace accordingly and avoiding pauses. This activity aims to reduce PWS’ dependence on auditory feedback, stimulating the fluent trajectory for phonatory acts.

During visual stimulation, participants are asked to synchronize the onset of their speech with an external visual cue, such as a clock that changes color on the screen. This cue indicates the precise moment to start speaking, thus providing an external timing cue. The timing for speech initiation varies continuously, preventing participants from predicting when to start the phonatory act. Participants are instructed to prepare for speech in a minimal activation state, thus reducing the typical effort associated with the phonatory act. In this activity, participants watch themselves on the monitor that zooms in on their lips or facial and body districts involved in the phonatory act. Even in this case, images have a slight and continuously adjusted delay. Visual stimulation has two primary objectives: first, participants learn to initiate speech under challenging conditions without anticipating the phonatory act, as determined by an external, unpredictable visual stimulus. In this way, they learn to be constantly ready to start speaking—as in everyday life. Secondly, participants experience and become aware of their reduced global activation during speech.

Visual stimulation may also be combined with somatosensory stimulation. In this case, participants are asked to wear a custom-made jacket with 32 sensors providing vibrations in various body regions (e.g., chest and lower abdomen), thus exaggerating the maladaptive hyperactivation experienced by PWS during stuttering. Even in this case, somatosensory stimulation is used as an external timing cue, increasing or reducing it when participants must initiate the phonatory act.

In individuals with severe stuttering, we generally introduce a non-phonatory motor act in this activity, such as allowing the patient’s arm to fall inertially from an initial position following gravity. This optional step aims at integrating multisensory stimulation with an external cue for the timing of speech and provides an example of a movement requiring minimal effort.

Acoustic, visual, and somatosensory stimulation occur consecutively on the first day. These activities are repeated to help participants achieving the target of fluency and as warm-up exercises to prepare them for the subsequent individual and small group activities. Once participants experience their natural voice, free of blocks and repetitions, fluent responses should be continuously sought and reinforced. Participants are asked to focus on the acoustic, muscular, respiratory, and proprioceptive information associated with the fluent voice and are instructed to replicate the same sensations during the phonatory act. The trainers record a short segment of the patient’s fluent spontaneous speech (approximately 20 s) to reinforce these acquisitions further. This recording serves as the new target. The patient’s produced speech is continuously compared to the recorded target through a digital process called Voice Live Feedback, in which they receive visual information considering voice frequency and amplitude compared to the target, facilitating the learning and reinforcement of new self-regulation trajectories to achieve the target voice. The time required for this activity varies among participants.

Throughout AMFS, the trainers tailor the protocol parameters and establish an intervention program based on each patient’s characteristics, providing a personalized multisensory intervention. Reading and speaking performance are continuously monitored during the treatment to adjust the intervention. Individual activities are conducted during the first two days, while the following three days include individual and small group sessions, typically involving 6–7 participants plus the trainer. In this phase, participants are exposed to feared or avoided situations (e.g., asking for information by telephone, speaking with strangers, etc.), which were reported by participants at the beginning of the intervention, and engaged in discussions with group members, fostering the consolidation and generalization of the acquired skills in a more ecological setting.

In the following section, we present data from a group of PWS who underwent AMFS. Stuttering severity was assessed before and after the intensive phase and during the reinforcement phase, with an average interval of three months after the end of the intensive phase. We recorded physiological parameters such as electromyographic activity, breath frequency, heart rate, peripheral temperature, and skin conductance to explore whether training effects influenced stuttering-associated behaviors.

#### Key Concepts—Glossary Used in the Introduction

*Stuttering response*: maladaptive and involuntary motor behavior in people who stutter (PWS). The stuttering response has been learned and reinforced through PWS experience and, therefore, is the automatic response when (internal or external) perturbations occur.

*Fluent response*: opposite to the stuttering response, it represents an effortless and fluent motor behavior. The fluent response is among the motor repertoire of PWS, who experience moments of fluent speech. However, it is less reinforced than the stuttering response in PWS and is not the primary choice in the case of perturbation.

*Effort*: here, we refer to the cognitive and physical effort observed in PWS when performing phonatory acts.

*Natural voice* is the participants’ voice as they experience it during moments of fluency, free from blocks and repetitions. Considering the re-education aim of AMFS, natural voice is not achieved through artificial signals or efforts, such as slowed speech, prolongations, or sing-song patterns.

## 2. Material and Methods

### 2.1. Participants

The PWS group comprised forty-six Italian native speakers (40 males, mean age = 25.9 ± 7.5), whereas the control group included 24 non-stuttering participants (22 males, mean age 32.5 ± 10.3). Data was collected between October 2022 and September 2023. The participants declared no history of neurological or psychiatric disorders nor current use of psychiatric medications. This study occurred at the Vivavoce Institute (Milan, Italy). All participants gave their informed consent before the study procedures.

### 2.2. Primary Outcome Measures: The Stuttering Severity Instrument

The Stuttering Severity Instrument Fourth Edition (SSI-4) [61] is a clinical tool that assesses stuttering severity based on three parameters: frequency, which is measured by the percentage of syllables stuttered; duration, which is measured by the three most extended stuttering events; and physical concomitants, such as loud breathing, facial grimaces, head, and body extremity movements. The parameters are based on clinical judgments of physical and audible signs of struggle during speech. The frequency percentage and duration scores are converted to a scale ranging from 2 to 18. Differently, physical concomitants are evaluated using a six-point scale (ranging from 0 = none to 5 = severe and painful looking). Scores from the three parameters can be summated to obtain a total score indicating the overall severity of stuttering, which can be classified into a severity rating scale (very mild, mild, moderate, severe, and very severe) [62].

### 2.3. Secondary Outcome Measures: Physiological Parameters

The physiological measures included heart rate as measured through blood volume pulse (BVP), electromyographic (EMG) mean amplitude, skin conductance response (SCR), peripheral temperature, and respiratory rate (number of breaths per minute). The physiological parameters were collected using the BioGraph Infiniti (version 6, Thought Technology Ltd., Montreal, QC, Canada), which recorded the parameters and provided the previously mentioned measures as outcomes. The physiological parameters were collected through a plethysmograph on the left hand’s third finger, two cutaneous conductance detectors on the second and fourth left fingers, and two stripes positioned around the thorax and the abdomen to control lung ventilation.

### 2.4. Experimental Procedure

In the PWS group, primary and secondary outcome measures were collected at three different time points: before starting the training (T0), immediately after the intensive phase (T1), and at follow-on, namely during the reinforcement phase (T2), two to four months after the end of the intensive phase. The outcome measures were collected only once for the control group, as these participants did not undergo the training since the original idea was to have a baseline comparison between PWS and the control group.

The baseline evaluation occurred on the first day of treatment before the start of the intervention. The participants sat in a quiet room with a camera positioned at a distance of 1.5 m. The camera was centered on the participants’ heads and torsos. The participants were then recorded (i) while introducing themselves, to collect a one-minute spontaneous speech sample, and (ii) while reading aloud a pre-selected text including at least 200 syllables. The trainers used the two recordings to perform an offline SSI-4 evaluation. The trainers were not blind to the participants’ group for the SSI-4 assessment.

After the recordings, sensors were placed on the participants to measure the physiological signals. The BVP sensor was placed on the non-dominant thumb, while the SCR sensors were attached to the non-dominant index and ring fingers. A temperature sensor was placed on the non-dominant index finger to measure peripheral temperature (ranging from 18 to 36 degrees Celsius). The respiration frequency was measured through a breathing belt. Two electrodes were placed on the sternocleidomastoid muscle to measure EMG activity. Once the sensors were placed, the participants were instructed to relax and rest their non-dominant hand (where the physiological sensors were attached) on their legs. They were asked to look at a central fixation point while a one-minute physiological resting-state recording was performed. Then, physiological parameters were collected while individuals introduced themselves and read aloud a short text—different from the previous one but of the same length. Finally, participants completed two questionnaires to identify each situation they usually avoid or are afraid of during their individual exposure activities.

The training was conducted as described in the previous paragraph. On the fifth and final days of the intensive phase, the T1 assessment was performed using the same procedure performed at baseline. The same evaluation was performed at follow-on (T2), which took place on average three months after the end of the intensive phase.

### 2.5. Statistical Analyses

PWS’ baseline characteristics were reported as median and interquartile range and frequency and percentage for continuous and categorical variables, respectively. Considering the primary endpoint, namely stuttering severity measured through the SSI-4, the effect of time (three levels: baseline, or T0; after the intensive phase, or T1; and at follow-on, or T2) was analyzed in the PWS group to measure the changes induced by the treatment. The stuttering severity score was computed by summing all non-zero variables that constitute SSI-4, indicating the presence of a stuttering condition. The effect of treatment was performed using non-parametric statistics, namely the Friedman test. Pairwise comparisons were performed using Conover’s all-pairs test, with Bonferroni correction applied for multiple comparisons. PWS’ SSI-4 scores at different time points were contrasted with the control group’s scores using the Mann–Whitney test for independent data.

Considering the secondary endpoint, i.e., physiological signals, we analyzed the effects of the interventions in the PWS group at the three different time points (T0, T1, and T2) and then compared PWS’ and the control group’s physiological outcomes in three different conditions, namely at rest, during spontaneous speech, and during a reading test. As for the primary endpoint, the analysis of the physiological output in the PWS group was performed using the non-parametric Friedman test. Pairwise comparisons were exclusively performed for significant comparisons using Conover’s all-pairs test, with Bonferroni correction applied for multiple comparisons. The pairwise comparisons between the PWS and controls were run using the Mann–Whitney U test with Bonferroni correction. All statistical analyses were performed using R statistical software version 4.0.4 [63] (https://cran.r-project.org/index.html, accessed on 3 August 2024).

## 3. Results

Considering the primary endpoint, namely stuttering severity measured through the SSI-4, pairwise comparisons on the total scores highlighted a reduction considering the severity score at T1 compared to T0 (*p* < 0.001) and at T2 compared to T0 (*p* < 0.001). In contrast, no differences emerged at T2 compared to T1 (*p* = 0.240) (see Figure 2).

We also analyzed the results in the specific components of SSI-4. For frequency, pairwise comparisons highlighted a reduction considering the frequency score at T1 compared to T0 (*p* < 0.001) and at T2 compared to T0 (*p* < 0.001). In contrast, no differences emerged at T2 compared to T1 (*p* = 1). For the duration, pairwise comparisons highlighted a reduction considering the duration score at T1 compared to T0 (*p* < 0.001) and at T2 compared to T0 (*p* < 0.001). In contrast, no differences emerged at T2 compared to T1 (*p* = 0.58).

For the physical concomitants, pairwise comparisons highlighted a reduction considering the physical concomitants score at T1 compared to T0 (*p* < 0.001) and at T2 compared to T0 (*p* < 0.001). In contrast, no differences emerged at T2 compared to T1 (*p* = 1).

Crucially, PWS and controls differed in their SSI-4 total scores at T0 (*p* < 0.001), whereas the differences between the two groups disappeared after the treatment (*p* = 0.127) and at follow-on (*p* = 0.800). The differences between the PWS and the control group in the SSI-4 subscales followed the same trend, highlighting worse performance in the PWS’ baseline than in the controls. Such differences disappeared in the PWS assessments after the intensive phase, except for the duration subscale, which showed a remaining trend toward significance (*p* = 0.071). Comparing performance at T2 with the baseline of the control group, no residual differences remained (all *p*s > 0.597) (see Appendix A for detailed comparisons).

Considering the secondary endpoint, namely physiological parameters, the results are separately presented for the rest condition, the spontaneous speech, and the reading conditions (for a detailed comparison, see Appendix A, respectively). Comparing PWS with the control group, no differences emerged at baseline in the resting state (all *p*s > 0.100) and during spontaneous speech (all *p*s > 0.118); the only difference between the two groups was in the reading condition, in which skin conductance was higher in the PWS than in the controls (*p* = 0.018).

Considering the treatment effect, in the resting-state condition, heart rate (*p* < 0.001) and mean temperature (*p* = 0.010) changed in the three measurements. Pairwise comparisons highlighted that heart rate decreased after the intensive phase and at follow-on compared to the baseline (*p* = 0.013 and *p* = 0.001, respectively). In contrast, no differences were found between T1 and T2 (*p* = 1), suggesting that the change remained stable between the end of the intervention and at the follow-on. The mean temperature increased at T2 compared to T1 (*p* = 0.011), whereas no difference was found at T1 and T2 compared to the baseline (*p* = 1 and *p* = 0.133, respectively). The comparison with the controls suggested higher temperature values in the PWS group, but only at T2 (*p* = 0.004).

In the spontaneous speech condition, BVP, EMG, and temperature changed during the treatment. Specifically, BVP decreased after the intensive phase and at follow-on compared to the baseline (*p* < 0.001). At the same time, the value remained stable between the end of the intensive phase and at follow-on (*p* = 1). The comparison with the controls suggested lower BVP in the PWS at T1 and T2 (*p* = 0.030 and *p* = 0.005, respectively). EMG activity also reduced during the treatment, with lower activation at T1 and T2 compared to the baseline (*p* < 0.001) and compared to the control group (*p* < 0.001). As for the resting-state condition, temperature increased at T2 compared to T1 in the control group (*p* = 0.042), highlighting a higher value than that in the control group (*p* = 0.004).

Finally, in the reading condition, BVP, EMG, and respiration rate decreased after the treatment (all *p* < 0.001). Specifically, BVP was lower at T1 and T2 than at T0 (*p* = 0.004 and *p* < 0.001, respectively), and remained stable between T1 and T2 (*p* = 0.444). Moreover, the heart rate at T2 was lower than that in the control group (*p* = 0.021). The EMG signal was lower at T1 and T2 than at T0 in the PWS group (*p* < 0.001) and remained stable between T1 and T2 (*p* = 1). EMG at T1 and T2 was lower in PWS than in the controls (*p* = 0.002 and *p* = 0.003, respectively). The respiration rate decreased at T1 and T2 compared to the baseline (*p* = 0.004 and *p* < 0.001, respectively) but remained stable between T1 and T2 (*p* = 0.120). Moreover, breath frequency after the intervention was lower than that in the controls (*p* < 0.001).

## 4. Discussion

The present work introduces AMFS, a novel method to treat PWS. The concept of AMFS was developed under the theoretical framework of the neurocomputational model DIVA [28,29] and its application to stuttering [30,31,33,34,64]. Specifically, the intervention addresses the speech motor control system’s over-reliance on sensory feedback, encompassing not only the auditory modality—which has been extensively explored in the literature (see, for a recent review, [30]) and in stuttering treatments (e.g., [43,65])—but also visual [45,46,47] and somatosensory feedback [48,49,50]. Our approach seeks to continuously perturb feedback controller inputs by providing multimodal stimulation involving acoustic, visual, somatosensory, and proprioceptive channels. We hypothesize that this exaggerated stimulation would disrupt the reliability of sensory feedback, leading to a functional breakdown, which is necessary to allow for the emergence of a fluent, instead of stuttering, response. The fluent response is PWS’s natural voice, as they experience it during moments of fluency, free from blocks and repetitions. The fluent response, therefore, is not achieved through artificial signals such as slowed speech, prolongations, or sing-song patterns, nor through muscular relaxation techniques. Instead, once participants’ natural voice emerges, we train them to become aware of acoustic, muscular, respiratory, and proprioceptive indexes experienced during natural speech, thus building stronger and stabilized motor plans during the reinforcement phase.

In the current work, we present preliminary evidence on the effectiveness of the AMFS intervention by measuring stuttering severity and physiological signals indicating secondary stuttering behaviors or general physiological or emotional activation (e.g., breath frequency, heart rate, and skin conductance measures) before (T0) and after an intensive phase (T1), as well as during the reinforcement phase or at follow-on (T2). The scores at the different time points have been compared with the performance of the control group based on the SSI-4 and the physiological indexes recorded.

The observed improvement encompasses all the parameters included in the SSI-4: frequency and duration of stuttering and physical concomitants. These findings are also strengthened when comparing PWS with the control group. At baseline, the two groups differ significantly in SSI-4 scores, with the PWS showing worse performance, whereas the control group does not show stuttering symptoms. However, the two groups do not differ in stuttering scores at the end of the intensive phase and at follow-on, thus highlighting comparable performance in stuttering and non-stuttering participants. Notably, the subscale on stuttering duration suggests that PWS maintained a ‘borderline’ performance at the end of the intensive phase compared to controls. This trend, however, disappears during follow-on, with PWS performing like the control group. This result has important significance, as it suggests that PWS not only improved during the intensive phase of the training but also continued to progress during the reinforcement phase, providing robust support for the rationale behind the AFMS intervention.

Previous works manipulating the different sensory modalities also suggest significant improvement in verbal fluency, typically reducing stuttering behaviors [43,45,46,50,66,67,68]. However, treatment maintenance has been rarely assessed [58] with studies suggesting that the effects are temporary and disappear when participants adapt to the perturbation or when the alteration of feedback ends [30,58,59]. Although preliminary, our results suggest that an intensive week of AMFS intervention can promote improvements that last even after the intensive phase, and future studies should clarify whether the beneficial effect can be measured even after the end of the reinforcement stage.

Our findings draw a complex picture considering the PWS physiological outcomes at the three time points. BVP was reduced after the intensive phase compared to baseline, and it remained stable at follow-on in all three recording conditions: resting state, spontaneous speech, and reading task. The EMG activity decreased after the intensive session recording and at follow-on compared to the baseline registration in the spontaneous speech and reading task. Finally, the respiration rate decreased only in the reading condition. Compared to the control group, PWS reported decreased BVP and EMG activity after the intensive phase and reinforcement during the spontaneous speech condition and in the reading task. During reading, respiration frequency decreased in PWS compared to the control group after the intensive phase and during the follow-on. Moreover, during this activity, a difference was found between PWS’ baseline and the control group for skin conductance, with PWS reporting higher levels than the control group.

Interpreting these findings is challenging, as previous studies produced heterogeneous results when exploring physiological patterns in PWS vs. non-stuttering individuals or exploring physiological responses during stuttered vs. fluent utterances [56,69,70,71]. The most consistent result in the literature is related to the heart rate decrease during speech in stuttering vs. non-stuttering participants [56,70,72,73]. This effect has been generally interpreted as a ‘freezing response’, namely an inhibitory anticipation reaction typically found in humans and mammals to unpleasant or fearful stimuli [56]. In PWS, freezing responses have been reported as difficulties in speech–motor initiation [74], likely reflecting dysfunctions in the basal ganglia loop. Similarly, Parkinson’s disease is characterized by freezing phenomena, manifesting as a sudden and temporary inability to initiate or sustain movements such as walking or speaking [75]. This overlap suggests that both conditions may share underlying neural mechanisms related to motor initiation deficits. Skin conductance and pulse volume previously yielded inconsistent results, with some research reporting no differences between stuttering vs. non-stuttering participants [72,73], while others have noted increased skin conductance in PWS during stuttered vs. fluent utterances [70]. Some research highlighted differences in respiratory patterns in PWS vs. non-stuttering participants, suggesting reduced coordination between respiratory, phonatory, and articulatory movements [69,71]. Studies investigating EMG activity also provided mixed results, with studies highlighting no differences in orofacial, mandibular, laryngeal, and respiratory muscular activity during stuttering [76,77,78], while others reported increased muscular activity during stuttering [41,79]. Considering our data, in many cases, PWS at T1 and T2 had lower physiological signals than the control group. This could be due to a limitation of our methodology, in which the control group had just a single exposure to the experimental setting and may feel more aroused than PWS. Contrary to this conjecture, no difference between PWS and the control group was found in the resting-state condition—the first condition assessed—where the only difference was in comparing the follow-on measure of temperature in PWS, higher than the control group.

Interpreting the results on physiological indices remains challenging, as the literature does not provide clear evidence on how these variables relate to speech fluency in both stuttering and non-stuttering individuals, nor whether their modulation directly benefits PWS symptoms. However, the observed reduction in EMG responses following the intervention, both in the spontaneous-speech and reading-aloud conditions, along with decreased respiration rate during reading aloud, aligns with previous findings suggesting reduced physical effort during speech production in PWS [37]. In contrast, the reduction in BVP was also present during the resting-state condition, possibly reflecting a general decrease in arousal due to task repetition rather than a fluency-specific effect. Future studies should further investigate the relationship between these physiological measures and fluency, helping to determine whether increases or decreases in these variables contribute to symptom improvement in PWS.

## 5. Limitations and Future Directions

This study has several limitations. First, while the control group was matched for gender, it was not matched for age and was assessed only once. This choice was made because the control group did not undergo any intervention and served as a baseline comparison for the PWS group. However, future studies should conduct multiple assessments in the control group to account for potential changes induced by the experimental paradigm. Additionally, our analyses of stuttering behaviors were limited to the SSI-4 parameters, without incorporating other relevant measures such as vocal response delay or self-assessment of stuttering’s impact on daily life. Instruments like the Overall Assessment of the Speaker’s Experience of Stuttering (OASES) [80] could provide valuable insights into the patient’s experience of stuttering, and the questionnaire is now part of routine clinical assessments at our center. Addressing these methodological limitations will be a priority in future studies.

Furthermore, future research should explore the underlying mechanisms of AMFS to refine the treatment approach, enhance methodological precision, and examine the protocol’s effectiveness beyond the reinforcement phase.

## 6. Conclusions

Taken together, our findings suggest that AMFS improved stuttering severity in PWS so that their performance was comparable to that of the non-stuttering controls at the end of the intensive session and, even more crucial, during the follow-up evaluation. The training also affected the physiological indexes, even though the direction of the pattern still needs to be clarified.

## 7. Patents

The technology applied for the AMFS has been recognized with a patent from the Italian Patent and Trademark Office (patent number: 102022000021813, De Vita & Muscarà, 2024).

## Figures and Tables

**Figure 1 brainsci-15-00246-f001:**
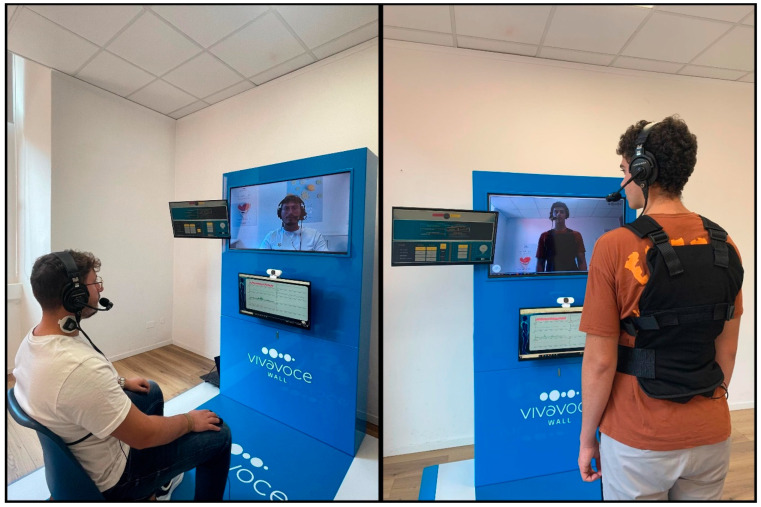
The figure illustrates some of the equipment used for AMFS. The **left** image shows a participant with biofeedback sensors during a fluency assessment session. The **right** image shows a participant during a rehabilitation session in the post-intensive phase, which includes auditory (microphone + headphones), visual (monitor), and vibrotactile (vest) stimulation.

**Figure 2 brainsci-15-00246-f002:**
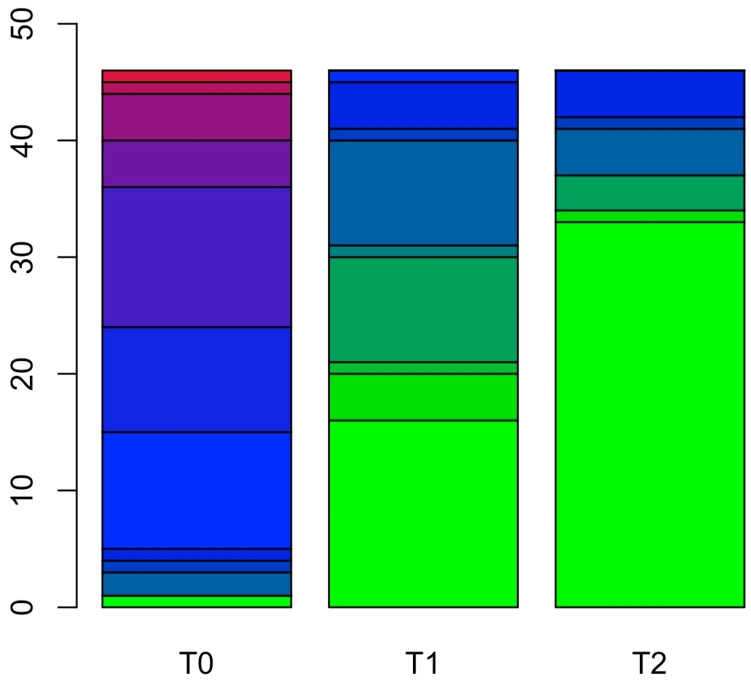
The figure depicts the different performance levels in SSI-4 total scores. The *y*-axis represents the number of participants at each time point (*x*-axis) and their levels of performance, which are depicted through different colors. Specifically, the color of the bar plot identifies a gradient of disease. The green color represents a disease score equal to 0, the shades of blue represent a disease score between 1 and 15 (lighter blue indicates a lower score, while darker blue indicates a higher score within the 1–15 range), and the red represents a disease score equal to 16.

## Data Availability

The original data presented in this study are openly available in OSF at this link: https://osf.io/quskp/, accessed on 16 January 2025.

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
