# Peer review of "Improving Stuttering Through Augmented Multisensory Feedback Stimulation"

_brainsci, 2025, doi:10.3390/brainsci15030246_

Round 1
Reviewer 1 Report
Comments and Suggestions for Authors
I enjoyed reading the paper, and I liked the topic. There is a good effort in this paper. However, some comments need to be considered.
1. The results section presents changes in physiological measures such as heart rate, electromyographic activity, and respiration rate, but I could not find a clear explanation of how these changes specifically contribute to fluency improvement.
2. Some measures decreased after treatment, while others remained unchanged, yet there is no theoretical or clinical justification for these mixed results. Please try to justify.
3. The study claims that AMFS led to significant improvement. Nevertheless, the control group was only tested once (baseline) and not followed over time. This means comparisons at T1 and T2 are limited in proving long-term efficacy. Your discussion must acknowledge that without a control group follow-up, it is unclear whether fluency improvements persist due to AMFS or other external factors. A brief discussion of this limitation should be added.
4. The results only report group-level statistics, but there is no mention of whether some participants responded better than others. If this is possible, try to address or justify it.
5. Given that stuttering severity varies significantly between individuals, I wonder if it is possible to analyze whether factors like initial severity, age, or gender influenced treatment outcomes. Did participants with more severe stuttering show different improvements compared to those with mild stuttering?.
6. Have a separate limitations section: limitations and directions for future research.
7. Have a separate conclusion section.
8. The paper should undergo professional language editing to ensure conciseness and clarity.
Comments on the Quality of English Language
The paper should undergo professional language editing to ensure conciseness and clarity.
Reviewer 2 Report
Comments and Suggestions for Authors
Thank you for the opportunity to review this manuscript.
The topic is of substantial interest, and I appreciated the thorough outline of the DIVA model.
However, one key component that the authors summarize is the possible role of the basal ganglia. In the background, the authors describe the basal ganglia only as part of a cortical loop, and while this may be an accurate description of the DIVA model assumptions, I fail to see how the basal ganglia's role in motor initiation can be fully disregarded in relation to stuttering.
A link between Parkinson's disease (PD) and stuttering has been seen/suggested and the author's observation of a freezing (L564) is congruent with some motor symptoms in PD. I am not suggesting that the authors should align very strongly to this possible link in their work, but I do think that the mechanisms of the freezing should be highlighted better.
P7 Please indicate whether the delay in vocal response can be measured from the data and the rationale for not measuring that parameter, especially considering a purported link to the initiation of motor actions highlighted above.
Please indicate whether the SSI-4 was performed blinded or not.
Please provide a rationale for not including a patient-reported outcome measure for stuttering outside of the clinical evaluation setting.
Please provide a rationale for not attempting to balance and match the control and experimental groups.
Figure 4: Please provide a key for thee color scale.
Text referring to Tables: Please refer to tables in supplementary materials in the text so the reader knows their purpose.
Comments on the Quality of English Language
L459 The results presentation contains many sentences with awkward formulations or repetitions. Please let a native academic speaker review them.
L 456 Please avoid this one-sentence paragraph.
Minor point:
In the Supplementary materials table captions, please change "boldly highlighted" to "highlighted in bold".
Round 2
Reviewer 2 Report
Comments and Suggestions for Authors
I am generally fine with the revisions made by the authors. Figure 2 does not, however, provide a clear presentation of the results. The colors used do not support a separation of disease scores. Further, the description of the figure in the figure caption is not accurate:
"Specifically, the green color represents a disease score equal to 0, the orange/yellow represents a disease score between 1 and 15, and the red represents a disease score equal to 16."
Please amend.
Author Response
Response Letter Reviewer 2
I am generally fine with the revisions made by the authors. Figure 2 does not, however, provide a clear presentation of the results. The colors used do not support the separation of disease scores. Further, the description of the figure in the figure caption is not accurate:
"Specifically, the green color represents a disease score equal to 0, the orange/yellow represents a disease score between 1 and 15, and the red represents a disease score equal to 16."
Please amend.
We thank the Reviewer for her/his comment. In the revised version of the manuscript, we modified the figure and amended the caption.